# Antidementia Effects of *Alternanthera philoxeroides* in Ovariectomized Mice Supported by NMR-Based Metabolomic Analysis

**DOI:** 10.3390/molecules26092789

**Published:** 2021-05-09

**Authors:** Charinya Khamphukdee, Orawan Monthakantirat, Yaowared Chulikhit, Chantana Boonyarat, Supawadee Daodee, Possatorn Aon-im, Juthamart Maneenet, Yutthana Chotritthirong, Prathan Luecha, Nazim Sekeroglu, Anake Kijjoa

**Affiliations:** 1Division of Pharmacognosy and Toxicology, Faculty of Pharmaceutical Sciences, Khon Kaen University, Khon Kaen 40002, Thailand; possa.torn@kkumail.com (P.A.-i.); prathanl@kku.ac.th (P.L.); 2Division of Pharmaceutical Chemistry, Faculty of Pharmaceutical Sciences, Khon Kaen University, Khon Kaen 40002, Thailand; oramon@kku.ac.th (O.M.); yaosum@kku.ac.th (Y.C.); chaboo@kku.ac.th (C.B.); csupawad@kku.ac.th (S.D.); juthamart_pp@hotmail.com (J.M.); 3Graduate School of Pharmaceutical Sciences, Khon Kaen University, Khon Kaen 40002, Thailand; yutthana_ch@kkumail.com; 4Department of Horticulture, Faculty of Agriculture, Kilis 7 Aralik University, 79000 Kilis, Turkey; nsekeroglu@gmail.com; 5ICBAS-Instituto de Ciências Biomédicas Abel Salazar and CIIMAR, Universidade do Porto, Rua de Jorge Viterbo Ferreira 228, 4050-313 Porto, Portugal

**Keywords:** *Alternanthera philoxeroides*, Amaranthaceae, Thai traditional medicine, ovariectomized mice, antidementia, β-amyloid aggregation inhibitor, flavones, metabolomics

## Abstract

The crude ethanol extract of the whole plant of *Alternanthera philoxeroides* (Mart.) Griseb was investigated for its potential as antidementia, induced by estrogen deprivation, based on in vitro antioxidant activity, β-amyloid aggregation inhibition and cholinesterase inhibitory activity, as well as in vivo Morris water maze task (MWMT), novel object recognition task (NORT), and Y-maze task. To better understand the effect of the extract, oxidative stress-induced brain membrane damage through lipid peroxidation in the whole brain was also investigated. Additionally, expressions of neuroinflammatory cytokines (IL-1β, IL-6 and TNF-α) and estrogen receptor-mediated facilitation genes such as PI3K and AKT mRNA in the hippocampus and frontal cortex were also evaluated. These effects were confirmed by the determination of its serum metabolites by NMR metabolomic analysis. Both the crude extract of *A. philoxeroides* and its flavone constituents were found to inhibit β-amyloid (Aβ) aggregation.

## 1. Introduction

Nowadays, aging of the population is increasing very steeply and the World Health Organization (WHO) estimates that 1.2 billion women will be at 50 years or over by the year 2030, which is nearly three times more than those of the year 1990. Every year, millions of women undergo profound physiological changes, also known as menopause, with a dramatic decrease in estrogen level [1]. Numerous studies reported that senile dementia-like behaviors in both female rodent studies and in clinical trial studies are associated with the lack of estrogen, which can be protected by early hormone replacement therapy [2]. Epidemiological studies also suggest that decreased concentration of estrogen after menopause may be responsible for a higher prevalence and greater severity of Alzheimer’s disease (AD) in women than in men [3]. AD is the most common form of dementia which is characterized by neurodegenerative and neuronal death from neurotoxicity, β-amyloid (Aβ) plaques, a product of the membrane-bound amyloid precursor protein (APP) cleaved by β-secretase. Brain inflammation is also a pathological hallmark of AD which is related to the intractable Aβ plaques that stimulate chronic inflammation [4]. Aggregated amyloid fibrils and neuroinflammatory cytokines, secreted by microglia and astrocytic cells, lead to neuronal dystrophy. This cascade induces reactive oxygen species (ROS) and inflammatory enzyme systems, leading to oxidative stress. Moreover, Aβ itself has been shown to act not only as a pro-inflammatory agent but also as a ROS generator. Thus, the intervention in the formation process of β-sheet and/or amyloid fibrils can be considered as an approach to halt or slow down the progression of AD [5,6,7]. With respect to the effect of estrogen on this parameter, the ovariectomized (OVX) mice model is considered as an appropriate model for mimicking human ovarian hormone loss that accelerates the menopausal process in women via deterioration of homeostasis [8]. Over the last decade, herbal medicine, plant-based functional food products, and numerous naturally occurring bioactive compounds have attracted much attention from researchers and medicinal specialists because herbal products such as flavonoids are less toxic and more effective for the prevention or treatment of aging-related diseases such as cardiovascular diseases, metabolic syndrome, cancer, menopausal symptoms and cognitive disorders [9]. Epidemiological studies reported that AD prevalence is lower in the East Asian population than in the Western counterpart [10]. Investigation on different dietary consumption revealed that consumption of fruits and vegetables rich in flavonoids was estimated to be 10–40 fold higher in East Asia when compared to Western countries [11]. Therefore, plants rich in flavonoids may be useful for the improvement of cognition not only for food supplements but also for drug development [2].

The decoction of the whole plant of *Alternanthera philoxeroides* (Mart.) Griseb (Family Amaranthaceae) is used in Thai traditional medicine for improvement of blood condition, stimulation of milk secretion and treatment of post-natal depression. The whole plant is also consumed as a vegetable for health tonic in Southeast Asia [12,13]. Moreover, the ethanol extract of *A. philoxeroides* is claimed for its capacity to improve menopausal depression due to its flavone constituents that exert effects on neurogenesis and neuroplasticity process via estrogen receptor-mediated reinforcement of cAMP-response element binding protein (CREB) and brain-derived neurotrophic factor (BDNF) gene transcription in the hippocampus and frontal cortex [14]. Additionally, these flavonoids also displayed an inhibitory effect on monoamine oxidases (MAO)-A and -B that influence the level of serological neurotransmitters [15]. The ethanol extract of leaf of *A. philoxeroides* was also reported to exhibit anti-inflammatory activity by stabilizing human red blood cell membranes [16]. Previous chemical studies of *A. philoxeroides* reported the isolation of several bioactive compounds including cytotoxic pentacyclic triterpene saponins philoxeroidesides A–D, antiviral and immunomodulatory pentacyclic triterpenes chikusetsusaponin IVa and calenduloside E, and MAOs inhibitor C-glycosylated flavone alternanthin and alternanthin B, flavone glycoside, *E*-propenoic acid-substituted flavone, torosaflavone E, demethyltorosaflavone D, luteolin 8-C-*E*-propenoic acid, chrysoeriol-7-*O*-rhamnoside, kaempferol as well as phenolic acids such as ferulic acid, salicylic acid, syringic acid, vanillic acid, *p*-hydroxybenzoic acid and chlorogenic acid [12,15]. Although polyphenols and flavonoids are non-essential nutrients, they contribute to an added-nutraceutical value to a diet. Additionally, natural health products, especially flavones, have received increasing attention due to their biological effects such as antioxidant, antimicrobial, anticancer, anti-inflammatory and neuroprotective activities [17]. For these reasons, we have investigated the antidementia activity caused by the hormone-modulated neuroprotective effect of flavone-rich *A. philoxeroides*. In order to validate our results, the NMR-based metabolomic analysis was used to investigate the effect of the ethanol crude extract of *A. philoxeroides* on dementia induced by ovariectomy.

## 2. Results

### 2.1. Screening for Radical Scavenging Effect of the Ethanol Crude Extract of A. philoxeroides

The radical scavenging activity of the ethanol extract of *A. philoxeroides* (AP extract) in vitro was determined by 2,2-diphenyl-1-picrylhydrazyl (DPPH) and 2,2-azino-bis(3-ethylbenzothiazoline-6-sulfonic acid) (ABTS) assays. For DPPH free radical scavenging assay, ascorbic acid was used as a reference antioxidant (IC_50_ = 2.4 ± 0.34 µg/mL). The AP extract showed a radical scavenging capacity with IC_50_ = 222.58 ± 0.080 µg/mL. For ABTS assay, the water-soluble α-tocopherol analogue, trolox, was used as a reference antioxidant. The AP extract showed the ability to scavenge the ABTS^•+^ radical with an IC_50_ = 384.00 ± 0.36 µg/mL, which was 7.5 times less than trolox (IC_50_ = 51.32 ± 0.24 µg/mL).

### 2.2. Screening for Cholinesterase Inhibitory Activity of the Ethanol Crude Extract of A. philoxeroides

The inhibitory activity of the AP extract on acetylcholinesterase (AChE) and butyrylcholinesterase (BuChE) was evaluated by Ellman’s method [18]. Tacrine, a selective AChE inhibitor, was used as a positive control. The IC_50_ and the selectivity index (SI = IC_50_ of BuChE/IC_50_ of AChE) were determined. The AP extract showed an inhibitory effect on AChE and BuChE with IC_50_ values of 2.06 ± 0.016 and 3.27 ± 0.011 mg/mL, respectively. The SI of the AP extract was 1.60, implying its partial selectivity toward AChE.

### 2.3. Effect of the Ethanol Crude Extract of A. philoxeroides on OVX-Induced Cognitive Deficits-Like Behavior

In order to evaluate whether estrogen deprivation (OVX mice model) induced cognitive dysfunction, three behavioral models were evaluated. Spatial cognitive performance relevant to reference memory, recognition memory and spatial working memory performance of OVX mice were performed in the *Morris water maze task* (MWMT), the novel *object recognition task* (NORT) and the Y-maze task, respectively. The MWMT revealed that the escape latency of each animal group was decreased by daily training, indicating the same degree of learning ability. The steady state of swimming time started from the fifth day (Figure 1A). Statistical analysis showed that the vehicle-treated OVX group had significantly deteriorated abilities to learn and remember the location of the hidden platform and to retrieve the platform location in the training test and the probe test, respectively, compared with the sham-operated animals, indicating the estrogen deprivation-induced reference memory deficits. Repeated hormone replacement therapy with 17β-estradiol (E2) (1 µg/kg/day) and the AP extract (250 and 500 mg/kg/day), in a dose-dependent manner, significantly ameliorated the acquisition and retrieval performance of reference memory diminished by OVX (Figure 1B) (for detailed statistical analysis, see Appendix A).

Similarly, the NORT also showed that the sham-operated group spent significantly more time exploring the novel object than the familiar object while the vehicle-treated control OVX mice failed to discriminate these two objects, indicating OVX-induced impairment of recognition memory. On the other hand, the OVX animals that received hormone replacement therapy with 17β-estradiol (1 µg/kg/day) and the AP extract (250 and 500 mg/kg/day) for 8 weeks significantly improved the discrimination performance in the test phase (Figure 2A) (for detailed statistical analysis, see Appendix A). In the Y-Maze task, significant effects of sham-operated and vehicle-treated OVX animals by decreasing a percentage of spontaneous alteration indicate sustained cognition. Both doses of the AP extract and 17β-estradiol (E2) (1 µg/kg/day) significantly increased the percentage of spontaneous alternation when compared to the vehicle-treated OVX group (for detailed statistical analysis, see Appendix A). In order to exclude the possibility of false positive of increasing locomotor activity by behavioral disinhibition that affects behavioral model test processes, mice were assessed for their locomotor activity by measuring the number of movements in the Y-Maze task. By counting total arms of entry, it was found that neither sham-operated nor OVX groups that received both 17β-estradiol (E2) and the AP extract altered the locomotor activity in the Y-Maze task.

### 2.4. Effect of the Ethanol Crude Extract of A. philoxeroides on OVX-Induced Lipid Peroxidation in the Brain

This experiment demonstrates that ovariectomy enhanced lipid peroxidation in the whole brain, as measured by the level of malondialdehyde (MDA), indicating a possible involvement of oxidative stress in cognitive deficits. Figure 3 showed a significant effect of estrogen deprivation on lipid peroxidation in the brain of OVX mice with an increase of the MDA level. For the treatment group, both the AP extract (250 and 500 mg/kg/day) and 17β-estradiol (E2) (1 µg/kg/day) had a significant effect on lipid peroxidation (with decreasing MDA value) when compared to the vehicle-treated OVX group (for detailed statistical analysis, see Appendix A). Therefore, it was concluded that the AP extract exerts an antioxidant effect by decreasing lipid peroxidation in the whole brain.

### 2.5. Effect of the Ethanol Crude Extract of A. philoxeroides on OVX-Induced Changes in Neuroinflammatory Cytokines and PI3K/AKT Pathway-Related Genes in mRNA Expressions of Mice Hippocampus and Frontal Cortex

Quantitative real-time PCR (QPCR) analysis was used to demonstrate possible pathways of AD. As shown in Figure 4, interleukin 1β (IL-1β), interleukin 6 (IL-6) and tumor necrosis factor α (TNF-α) mRNA expressions were significantly increased in both hippocampus and frontal cortex of the OVX mice when compared with those in the sham-operated group. Interestingly, administration of 17β-estradiol (E2) and the AP extract, at 250 and 500 mg/kg/day, to the OVX mice decreased IL-1β, IL-6 and TNF-α mRNA expressions. When compared to the sham-operated group, expressions of PI3K and AKT mRNA in the OVX mice were significantly decreased in both brain regions. However, the expressions of these genes were reversed by treatment with both 17β-estradiol (E2) and AP extract, which normalize the down-regulated expressions of PI3K and AKT mRNA in OVX mice (for detailed statistical analysis, see Appendix A).

### 2.6. Serum Metabolic Profile of the OVX Mice Treated with the Crude Ethanol Extract of A. philoxeroide

^1^HNMR-based untargeted metabolomics was performed to investigate metabolic alteration in the serum of mice administered with either deionized water (DI), 17β-estradiol (E2) or the AP extract (250 and 500 mg/kg/day). Metabolites were identified by median spectra as shown in Figure 5 and Appendix A.

To visualize metabolic similarities and differences among groups, principal component analysis (PCA) model consisting of all samples was constructed with R^2^ (PC1 = 43.0% and PC2 = 16.7%) and Q^2^ = 0.648 (Figure 6A). High dose (500 mg/kg/day) of the AP extract-treated group was separated from the other groups as determined by PC1. The pairwise PCA model whose R^2^, with PC1 = 46.3% and PC2 = 16.5%, and Q^2^ = 0.366 between the high dose of the AP extract-treated group and the control group (DI water) showed a clear class separation, indicating certain metabolic alteration (Figure 6B). However, both low doses of the AP extract (250 mg/kg/day) and 17β-estradiol (E2) were much closer to the control group (DI water), indicating that OVX mice treated with a high dose of the AP extract presented the best performance against the deviations induced by OVX (Appendix A).

In order to identify biomarkers contributing to such a serum metabolic alteration in OVX mice treated with high dose of the AP extract, orthogonal projections to latent structures-discriminant analysis (OPLS-DA) model was constructed (R2X = 61.1%, Q2Y = 79.1%, permutation *p*-value = 0.01). The OPLS-DA score plot (Figure 6C) exhibited a maximum separation between the control group and the high dose of the AP extract-treated group. OPLS-DA discriminatory metabolites obtained from the loading plots were summarized in Appendix A. We have found that serum levels of tyramine, betaine, propylene glycol, α-D-glucose and glycerol were elevated in the high dose of the AP extract-treated group (Appendix A). Therefore, five differential metabolites were selected as candidate biomarkers associated with the OVX mice treated with a high dose of the AP extract.

To identify the metabolomic pathway of metabolites from the serum of the OVX mice treated with high dose of AP extract, Metabolic Pathway Analysis (MetPA) was used to perform the most relevant metabolites (tyramine, betaine, propylene glycol, α-D-glucose and glycerol) and metabolic pathway. The summarized pathway analysis (Figure 7) revealed that these metabolites were significantly responsible for only the galactose metabolism pathway (*p* < 0.05) (Appendix A).

### 2.7. Amyloid Aggregation Inhibition Effect of the Crude Ethanol Extract of A. philoxeroides and Its Flavone Constituents

The key hallmark of AD pathogenesis is the formation of toxic Aβ plaques in the brain of AD patients. Therefore, preventing or reducing the aggregation of Aβ has been a primary goal of several therapeutic strategies under development or in clinical trials. In the present study, the AP extract (100 µg/mL) and its constituents, alternanthin (**1a**), alternanthin B (**1b**), chrysoeriol 7-rhamnoside (**2**), torosaflavone E (**3a**) and demethyltorosaflavone D (**3b**) (Figure 8), at a concentration of 100 µM, were evaluated for self-induced Aβ1-42 aggregation using thioflavin T (ThT) fluorometric assay with curcumin (**4**), at a concentration of 5 µM, as a reference compound. The results showed that **4** could inhibit the aggregation of Aβ with a percentage of inhibition = 51.01 ± 3.93. All the test compounds exhibited good activity with a percentage of inhibition ranging from 51 to 82 µM (Figure 9). Compound **1b** was the most potent, exhibiting the percentage inhibition value of 81.96 ± 2.14 whereas the AP extract showed the percentage inhibition value at 83.25 ± 4.25.

## 3. Discussion

The present study investigated the effects of the AP extract on senile dementia-related behaviors using OVX mice model as well as the underlying molecular mechanism. Estrogen deprivation in menopause is suggested to be related to a high risk for development of AD, however, there is still a debate whether estrogen and phytoestrogen replacement therapy could result in reducing the risk of senile dementia of Alzheimer’s type in menopausal women [19]. For this reason, we chose the OVX mice model to study the senile dementia-related behaviors as well as to prove the effect of 17β-estradiol (E2). This model is widely considered as the best tool to mimic women with ovarian hormone loss, being able to cause a premature aging and inflammation of neurons and oxidative stress [20]. On the basis of these pathological alterations in the central nervous system (CNS), the present findings demonstrated that the AP extract could prevent senile dementia in menopausal women induced by estrogen deprivation. Moreover, the untargeted NMR-based metabolomic analysis revealed the metabolites profile as the primary bioinformatics data of the blood serum of the OVX mice, treated with the AP extract, confirming that this plant extract exerts its effect on the brain of the OVX mice. Moreover, flavones, isolated from *A. philoxeroides*, are believed to provide a shield mechanism against Aβ aggregation. This prompted the need for further studies of such diverse biological activities for neuroprotection [17].

It is well established that 17β-estradiol (E2) is a major female ovarian hormone whose deficiency may contribute to possible damage to the brain that triggers senile dementia in menopausal women [3]. The present study revealed that administration of the AP extracts improved cognitive deficits-like behavior in OVX mice as demonstrated by their behaviors in the MWMT, the NORT and the Y-maze task, respectively. We have found that estrogen depletion induced by ovariectomy was an important pathogenic factor that impaired the cognitive function in OVX mice. Both the AP extract and 17β-estradiol (E2) significantly enhanced a non-spatial cognitive performance in OVX mice observed in the NORT as well as a spatial cognitive performance observed in the Y-maze test. In the Y-maze test, which is widely used for short-term memory to measure the willingness of mice to explore new environments, mice regularly prefer to investigate a new arm of the maze rather than returning to the one that was previously visited [21]. In order to confirm the results obtained from the two experiments, the MWMT was also performed. This model is based on spatial working and reference memory, the capacity to keep spatial information active in working memory in a short period which is strongly related with *N*-methyl-D-aspartate (NMDA) receptor and hippocampal synaptic plasticity function, a key proceeding responsible for structural changes that occur in neurons during learning and memory formation [22,23]. In this study, the sham-operated mice did not show any difference in spatial working memory between the training and the test periods whereas the vehicle-treated OVX mice failed to discriminate learning and recall memory in the test phase. These results indicate that lack of estrogen that caused an impairment of a long-term spatial working memory involved synaptic plasticity-related signaling in the hippocampus and frontal cortex. Surprisingly, OVX mice treated with the AP extract showed an ability to recover from spatial working memory deficits induced by ovariectomy, similar to treatment with 17β-estradiol in the MWMT. According to the results obtained from behavioral studies linked with learning and memory, it is likely that the AP extract has a capacity to improve non-spatial, short-term spatial and long-term spatial working memory impairment induced by estrogen deprivation. Moreover, all of the behavioral experiments had no effect on locomotor activity indicating that the AP extract did not stimulate the CNS which may be related to a possibility of producing a false positive in activity associated with behavioral tests [24].

In order to better understand the effect of 17β-estradiol (E2) and the AP extract on cognitive dysfunction induced by ovariectomy in behavioral studies, oxidative stress-induced brain membrane damage was investigated. Evidence indicates that estrogen deprivation is a risk factor for brain damage, leading to neuronal death. The possible mechanism is the up-regulation of microglial proteasome activity by estrogen through the p42/44 mitogen-activated protein kinase (MAPK) pathway, which is vital for a rapid and efficient turnover of oxidized or otherwise damaged proteins and thus maintains microglial homeostasis in response to oxidative stress and inflammation [25]. Furthermore, neuroprotective effects of antioxidant compounds and phytoestrogen have been shown to reduce neuronal death by oxidative stress in the brain via lipid peroxidation process [26]. Consistent with the previous findings, thiobarbituric acid reactive substances (TBARs) levels, the biomarker of lipid peroxidation, in the whole brain significantly increased in the OVX mice compared with the sham-operated mice, and this elevation was blocked by the AP extract in a dose-dependent manner. Interestingly, the AP extract was found to exhibit a better suppression of lipid peroxidation than 17β-estradiol (E2) in the whole brain of OVX mice. These findings suggest that OVX-induced oxidative stress via estrogen deprivation is severe enough to cause membrane damage in the brain, leading to cognitive dysfunction. Since 17β-estradiol (E2) has a phenolic hydroxyl group in its structure that seems to inhibit oxidative stress via lipid peroxidation [27], it is suggested that the inhibition of lipid peroxidation by the AP extract is due to the phenolic hydroxyl groups of the flavonoid constituents of *A. philoxeroides* [12,15,28]. In agreement with the previous findings, our results showed that the AP extract exhibited a strong free radical scavenging activity in both DPPH and ABTS assays.

Numerous studies indicated that menopause favors a production of free radicals-modulated oxidative stress which is associated with neuroinflammation, elevated cytokine levels in many brain regions causing altered neurogenesis, hippocampal synaptic malfunction and cognitive impairment [25,29]. Therefore, we also investigated whether 17β-estradiol (E2) and the AP extract directly regulate neuroinflammatory processes in the brain. The results showed that lack of ovarian hormone significantly induced up-regulation of neuroinflammatory cytokines mRNA expressions, including IL-1β, IL-6 and TNF-α, which are found in neurons and glia. These findings are consistent with clinical reports of the overexpression of IL-1β, IL-6 and TNF-α. Tellingly, IL-1β and IL-6 in the brain of AD patients restrain the function of cholinergic systems and activate Aβ aggregation, leading to the accumulation of neurofibrillary tangle (NFT). On the other hand, TNF-α is a key element in inflammatory cascade and increases the Aβ and tau which are related to the pathogenesis of AD [7]. Interestingly, both 17β-estradiol (E2) and the AP extract (250 and 500 mg/kg/day) were found to significantly up-regulate all neuroinflammatory cytokines in both hippocampus and frontal cortex of the OVX mice. To confirm these results we investigated the estrogen receptor-mediated reinforcement of PI3K and AKT genes transcription which play the key role in neuronal survival, restoration of neuronal damage and modulation of neuroinflammatory response [30]. Our data revealed that ovariectomy down-regulated the expression of PI3K and AKT mRNA in both brain regions, however, this was improved by treatment with 17β-estradiol (E2) and the AP extract. This observation is in agreement with Fan et al. [31] who found that 17β-estradiol (E2) is necessary for object-recognition memory enhancement via activation of dorsal hippocampal PI3K/Akt and extracellular signal-regulated kinases (ERK) signaling pathways in young and middle-aged female but not in aged female mice. Several studies have shown that chronic neuroinflammation in the brain increases the risk for AD. Previous studies suggested that lack of ovarian hormone in OVX animals can also contribute to neuroinflammation by increasing IL-1β, IL-6 and TNF-α via PI3K/AKT-mediated pathway by decreasing PI3K and AKT genes transcription. In this context, *A. philoxeroides* can be considered as a potential candidate for neuroprotection as a number of flavonoid-rich plants have been proved to possess estrogenic activity that can ameliorate neurodegeneration and neuronal death.

In an attempt to determine if the AP extract exerts its effect on the brain, we analyzed the serum metabolites of all interventions in the OVX mice using a score plot for PCA of ^1^H NMR spectra. A total of 28 metabolite biomarkers were identified (Figure 5). However, only the high dose of the AP extract (500 mg/kg/day) showed significance when compared with the vehicle-treated animal, revealing a clear separation by the OPLS-DA (Figure 6C). This phenomenon may be due to a synergistic effect of chemical constituents, especially flavonoids of *A. philoxeroides*, that resulted in a better activity than that of 17β-estradiol (E2) (1 µg/kg/day). However, a higher dose of 17β-estradiol (E2) should be further investigated to clarify if it also strongly affects senile dementia-like behaviors. The overall different metabolites between the AP extract (500 mg/kg/day)-treated and vehicle-treated groups allowed us to predict the pathways that are affected by the AP extract. MetPA analysis, in conjunction with Human Metabolome Database, revealed that the AP extract interferes with the galactose metabolism pathway. It is important to point out that decreased glucose supply for brain cells is related to the onset of AD since, in brain cells, this deficit leads to a decreased concentration of UDP-glucose and consequently of its 4′-epimer UDP-galactose as available precursors of structural components [32]. Moreover, chronic oral galactose administration was found to prevent the development of cognitive deficits in an animal model by stimulation of endogenous glucagon-like peptide 1 (GLP1)-mediated normalization of cerebral glucose hypometabolism, thus ameliorating neurodegenerative disorder [33]. Our results showed that the AP extract did not significantly exert its effect on both cholinergic neurotransmitters, AChE and BuAchE, which is in line with the fact that choline and amine neurotransmitters play an important role in regulating many processes, including memory, learning, mood, and behavior. Since *A. philoxeroides* and its flavone constituents have been reported as inhibitors of MAO, the enzyme principally responsible for the degradation of amine neurotransmitters [15], we suggest that betaine, commonly found in the *Amaranthaceae* family, is a potential biomarker metabolite of the extract of *A. philoxeroides* [12].

In order to ensure whether flavonoids could be responsible for the antidementia effect, we also investigated the β-amyloid aggregation inhibitory effect of the AP extract and its flavone constituents *viz.* alternanthin (**1a**), alternanthin B (**1b**), chrysoeriol 7-rhamnoside (**2**), torosaflavone E (**3a**), and demethyltorosaflavone D (**3b**) (Figure 8) in the present study. The accumulation of β-amyloid plaques is considered as a key hallmark of AD pathogenesis through an inflammatory cascade in the brain via overproduction of free radicals and cytokines such as IL-1β, IL-6 and TNF-α [4]. The results showed that the AP extract and the five flavone derivatives exhibited a potent inhibitory effect on Aβ aggregation, ranging from 53 to 84% (Figure 9). Interestingly, **1b** exhibited the most potent activity and even more potent than curcumin (**4**), with 82% of inhibition. This is not surprising because many flavonoids have been previously reported to possess a significant inhibitory effect on Aβ aggregation [34]. On the other hand, our previous study on the effect of these five flavone derivatives on the function of monoamine oxidase (MAO) enzymes [15] showed that they exerted the MAO-A and MAO-B inhibitory effects. While **1a** exhibited a more potent inhibitory effect on MAO-A when compared with clorgyline (a selective MAO-A inhibitor), **1b** showed a more potent inhibitory effect on MAO-B when compared with deprenyl (a selective MAO-B inhibitor). It is well established that MAOs play an important role in several key pathophysiological mechanisms in AD and neurodegenerative diseases. MAO-B has been proposed as a biomarker since activated MAO-B can lead to cognitive dysfunction, destroys cholinergic neurons, causes disorder of the cholinergic system and contributes to the formation of Aβ plaques. Moreover, MAO inhibitors also improve cognitive deficits and reverse Aβ pathology by modulating proteolytic cleavage of amyloid precursor protein, thus decreasing Aβ protein fragments. Taken together, these findings confirm that the pharmacological activity of these flavone derivatives is responsible for the antidementia effect of the AP extract.

## 4. Experimental Section

### 4.1. Plant Material, Preparation of Plant Extract and Isolation of Flavone Constituents

Collection, identification and isolation of secondary metabolites from the ethanol extract of *A. philoxeroides* (Mart.) Griseb have been described previously by Khamphukdee et al. [15]. The crude ethanol extract was freeze-dried into powder before giving to animals.

### 4.2. Radical Scavenging Activity

#### 4.2.1. DPPH Radical Scavenging Assay

DPPH radical scavenging assay was performed using the previously described method [35]. Ascorbic acid (Sigma-Aldrich, St. Louis, MO, USA) was used as a standard. The crude extract was aliquoted to 1 mL and mixed with 2 mL of 0.15 mM DPPH (Sigma-Aldrich, St. Louis, MO, USA). The mixture was gently mixed and left in the dark at room temperature for 30 min after which the absorbance was measured at 515 nm. The percentage of scavenging activity was calculated as follows:% scavenging activity = [1 − (At/Ai)] × 100
where A_i_ is the absorbance of the control and A_t_ is the absorbance in the presence of the test compound.

#### 4.2.2. ABTS Radical Scavenging Activity Assay

The ABTS radical scavenging activity assay was performed according to the method previously described by Thiratmatrakul et al. [36]. ABTS (3.6 mg) (Sigma-Aldrich, St. Louis, MO, USA) and K_2_SO_8_ (0.67 mg) were dissolved in 1 mL of distilled water. ABTS^+^**^.^** was generated and left in the dark at room temperature for 12 h. This solution was diluted with ethanol. Then, 10 µL of ABTS^+^**^.^** and 990 µL of the plant extract or reference compound, Trolox (Sigma-Aldrich, St. Louis, MO, USA), were transferred to a 96-well plate and mixed, and the absorbance was measured at 734 nm. The percentage of scavenging activity of the extract or standard was calculated as follows:% scavenging activity = [(A_ABTS_^+^**^.^** − A_t_)/A_ABTS_^+^**^.^**] × 100
where A_ABTS_^+^**^.^** is the absorbance of ABTS^+^**^.^** and A_t_ is the absorbance in the presence of the test compound.

### 4.3. Cholinesterase Inhibition Assay

Cholinesterase (ChE) inhibition was determined by Ellman′s method [18] in 96-well microplates. 50 µL of a sample solution was mixed with 125 μL of 1 mM DTNB (5,5-dithio-bis(2-nitrobenzoic)acid) (Sigma-Aldrich, St. Louis, MO, USA) and 25 μL of AChE (Sigma-Aldrich, St. Louis, MO, USA) or BuChE (Sigma-Aldrich, St. Louis, MO, USA) solution in Tris–HCl buffer (pH 8.0). The mixtures were incubated at 25 °C for 15 min. The reaction started with the addition 25 μL of 1 mM acetylthiocholine iodide or butyrylthiocholine iodide (Sigma-Aldrich, St. Louis, MO, USA) as substrates. The hydrolysis of acetylthiocholine iodide was measured by the formation of the colored product, 5-thio-2-nitrobenzoate anion, which is formed by the reaction of DTNB and thiocholine. Tacrine (Sigma-Aldrich, St. Louis, MO, USA) was used as a positive control. The absorbance was measured at 405 nm by UV/visible spectrophotometer every 30 s over the period of 5 min. The results were expressed as 50% inhibition (IC_50_) of ChE. The IC_50_ was calculated graphically from a concentration–inhibition curve [36].

### 4.4. Animals

Fifty female ICR mice were obtained at the age of 5 weeks from the National Laboratory Animal Centre at Nakhon Pathom, Thailand. The animal experimental procedures used in this study were in accordance with the Guiding Principles for the Care and Use of Animals (NIH Publications No. 80-23, revised in 1996). The animals were habituated in a light-controlled room with a 12-h dark/light cycle, under controlled temperature (22 ± 2 °C) and humidity (45 ± 2%). Food and water were freely available in the Laboratory Animal Unit of the Faculty of Pharmaceutical Sciences, Khon Kaen University, Thailand. The present studies were approved by the Animal Ethics Committee for Use and Care of Khon Kaen University, Thailand (IACUC-KKU-103/60).

### 4.5. Surgical Operation

Ovariectomy was conducted as previously described by Khamphukdee et al. [15]. Female mice were anesthetized with Nembutal^®^ (Ceva Sante Animal, Los Altos, CA, USA) (60 mg/kg, *i.p*.) and bilaterally ovariectomized. An incision (1–3 cm) was made at cranial terminus 1 cm caudal to the 13th rib. Skin incision was closed by suture and healed by antiseptic solution. The sham-operated group was performed in a similar procedure but without ovariectomy.

### 4.6. Drugs and Drug Administration

All treatments were performed three days after ovariectomy. The animals were divided into 6 groups: (1) sham-operated and (2) OVX groups, both of which were administered with distilled water (vehicle, per oral (*p.o*.)), (3) OVX + E2 (1 µg/kg of 17β-estradiol, *p.o.*), (4) OVX + AP250 (AP extract 250 mg/kg/day, *p.o*.), and (5) OVX + AP500 (AP extract 500 mg/kg/day, *p.o*.). 17β-Estradiol (Sigma-Aldrich, St. Louis, MO, USA) and the AP extract (1 g) were suspended in the vehicle (10 mL) and intragastrointestinally administered once daily at doses 250 and 500 mg/kg/day (in total volume 0.2 mL) per mouse for 8 weeks (Figure 10). The behavioral experiments started 6 weeks after ovariectomy. 17β-Estradiol and the AP extract were administered 1 h before behavioral tests. After finishing the behavioral studies, mice were anesthetized with Nembutal^®^ (Ceva Sante Animal, Los Altos, CA, USA) 60 mg/kg, *i.p*. Blood serum was collected by cardiac puncture and centrifuged (3000 rpm; 4 °C) for 15 min for NMR-metabolomic analysis. The whole brain was collected to study lipid peroxidation whereas the hippocampus and frontal cortex were collected to determine pro-inflammatory cytokines mRNA expression. All tissues were quickly dissected out and kept at −20 °C for all experiments. Figure 10 shows a schematic presentation of animal experimental design.

### 4.7. Behavioral Tests

#### 4.7.1. Novel Object Recognition Test (NORT)

The NORT was investigated for learning test for neurobiological studies. The apparatus was made by a black matt polypropylene box (65 cm long, 45 cm wide and 45 cm tall). Two pairs of objects were randomly selected from a set of four objects and each object differs in shape, surface color, contrast and texture. The different objects were: two transparent glass round bottles filled with black water (5 cm diameter, 12 cm tall), two transparent plastic cuboid bottles filled with green water (8 cm diameter, 15 cm height). The two pairs of objects were rotated over the test session. The experiments consisted of two sessions as a sample trial (T1) and a test trial (T2). Twenty-four hours before experiments, a habituation test was performed. Mice were placed individually in the apparatus without any object and environment of the experimental room for 5 min to familiarize with the box. In T1, two similar objects were placed in symmetrical positions in the box. The center of each object was positioned 24 cm from the box wall. Mice were placed facing the wall and let them explore the objects including 5 cm diameter area for 5 min. T2 was conducted 30 min after T1 one of the two objects was replaced with a new object (object N). The effect of exploration and discrimination of mice was observed for 5 min by a video camera mounted above the apparatus and monitored on the computer by an observer located in another zone of a quiet experimental room. Memory was assessed by measuring the ability of mice to recognize the object previously present (object F). The time spent exploring the familiar object during 5 min was calculated as the percentage of discrimination index [(N − F)/(N + F)] × 100 [37].

#### 4.7.2. Y-Maze Task

Immediate working memory performance is assessed by recording spontaneous alternation. It is a behavioral test for the determination of the willingness of mice to survey new environments. Mice prefer to explore a new arm of the maze rather than repetition to that previously entered [38]. The Y-maze consists of three arms of equal size. Each arm has a V-shape corridor made of black matt polypropylene. The arms were 38.5 cm long, 3 cm wide and 13 cm tall, and were oriented at 60° angles from each other. The area was cleaned using 70% ethanol between trials in order to avoid odor trails. Mice were placed in the single trial at the end of one arm and were allowed to freely explore the Y-maze for 8 min. Number and sequence of arm visits were recorded manually by the observer. The entry was considered to be completed when the hind paws of the mouse have completely entered the arm. Alternation was defined as successive entries into the three different arms. Percentage alternation was calculated as the ratio of actual to possible alternation (defined as the total number of arm entries minus two), multiplied by 100 as shown in the following formula :% Alternation = [(No. of alternations)/(Total arm entries − 2)] × 100.

#### 4.7.3. Morris Water Maze Task (MWMT)

The MWMT is used to evaluate spatial memory and working memory by recording the escape latency and time spent in the quadrant. The water maze was a circular pool (65 cm diameter, and 25 cm deep), filled with water (25 ± 1 °C). The pool was divided into four quadrants. An escape platform was placed in the middle of one quadrant, 1.0 cm below the water surface, equidistant from the sidewall and in the middle of the pool. The platform that provides the only escape from the water was located in the same quadrant in every trial. Three different starting points for mice were placed around the perimeter of the pool. The training period continued for 5 days to reach the steady state of escape latency and exclude the bad memory mice. Mice were placed in the pool of water and allowed to swim freely for 60 s, 4 trials per day. The time that mice escaped and reached the platform was recorded. On the sixth day, the platform was taken off from the pool and mice were allowed to swim for 60 s per quadrant. The trained mice escaped and swam in the quadrant where the platform was formerly located. The swimming time at the quadrant where the platform used to be located was recorded as a measure of the ability of learning and memory [26].

#### 4.7.4. Locomotor Activity Test

In order to exclude false positive results from animal behavioral tests, the locomotive activity test was conducted by evaluation of mice by the Y-Maze task and the total arm entries were observed after 8 min.

### 4.8. Lipid Peroxidation

Lipid peroxidation by free radicals causes a degradation of lipid that induced cell damage, especially in the brain. Malondialdehyde (MDA) is a secondary product of this process that was estimated by measuring the TBARs levels [21]. Lipid peroxidation was measured in the homogenates obtained by homogenizing the whole brain with 10 volumes of the ice-cold phosphate buffer (5 mM, pH 7.4). The homogenized brain (10% *w*/*v*) was mixed with trichloroacetic acid (Sigma-Aldrich, St. Louis, MO, USA) and centrifuged at 8000× *g* at 4 °C for 10 min. The supernatant of the homogenized brain was incubated with 0.8% (*w*/*v*) of 2-thiobarbituric acid (Sigma-Aldrich, St. Louis, MO, USA) at 100 °C for 15 min. After cooling, the TBA-MDA complex was determined by measuring absorbance at 532 nm by UV/Vis spectrophotometer. MDA (Sigma-Aldrich, St. Louis, MO, USA) was used as a reference standard. The results were represented as ρmol of MDA/mg protein.

### 4.9. Protein Determination

Protein was determined in the homogenized brain by Bradford’s method using bovine serum albumin (BSA) as a reference standard (Sigma-Aldrich, St. Louis, MO, USA). 2 µL of homogenized brain was mixed with 25 µL of distilled water. 200 µL of Bradford dye reagent (Bio-Rad, Hercules, CA, USA) was added and mixed homogeneously. The absorbance was measured at 595 nm [39].

### 4.10. Quantitative Real-Time PCR (qPCR)

Mouse proinflammatory cytokines, IL-1β, IL-6 and TNF-α, and the estrogen receptor-mediated PI3K AKT mRNAs expressions in the hippocampus and frontal cortex were quantified by real-time PCR. Total RNA was extracted from the tissues with TRIzol^®^ (Thermo Fisher Scientific Inc., San Jose, CA, USA) according to the manufacturer′s instructions. First-strand cDNA was synthesized with oligo (dT) primers and SuperScript III reverse transcriptase (Thermo Fisher Scientific Inc., San Jose, CA, USA). The qPCR was conducted using SsoAdvanced™ Universal SYBR Green Supermix (Biorad, Hercules, CA, USA). The following primers were synthesized by Macrogen (Seoul, South Korea): Amplification was carried out using gene-specific PCR primer sets as follow: (1)-β-actin: 5′-AAC GGT CTC ACG TCA GTG TA-3′ (sense) and 5′-GTG ACA GCA TTG CTT CTG TG-3′ (antisense); (2)-IL-1β: 5′- GAC AGC AAA GTG ATA GGC C-3′ (sense) and 5′-CGT CGG CAA TGT ATG TGT TGG-3′ (antisense); (3)-IL-6: 5′-CTT CCA TCC AGT TGC CTT CTT G-3′(sense) and 5′-AAT TAA-3′ (antisense); (4)-TNF-α: 5′-GCC TCT TCT CAT TCC TGC TTG-3′ (sense) and 5′-CTG ATG AGA GGG AGG CCA TT-3′ (antisense); (5)-PI3k:-5′-GTG TCA GCG CTC TCC GCC 3′ (sense) and 5′ CTG ATA ATT GAT GTA TGG 3′ (antisense); (6) AKT:-5′ GTG TC CAG TGT AGA ATG ACT C 3′ (sense) and 5′ ATC TGT CGG AGA ACA CAC ATG 3′ (antisense). A melting curve analysis of each gene was performed each time after amplification. β-Actin mRNA was used as a control to which the results were normalized. Fold difference relative expressions were calculated.

### 4.11. Determination of Untargeted Metabolites by NMR-Metabolomic Analysis

Serum metabolomic profiling was conducted at Khon Kaen University International Phenome Laboratory (KKUIPL). Serum samples were defrosted and spun at 1800 g, 4 °C for 10 min. Then, 300 µL of supernatant was transferred to a new Eppendorf tube and mixed with 300 µL of serum buffer pH 7.4 containing 0.075 M Na_2_HPO_4_, 2 mM NaN_3_ and 0.08% sodium trimethylsilyl-[2,2,3,3-^2^H_4_]-propionate [TSP]. The mixture was briefly spun down and 580 µL of supernatant was transferred to the NMR tube (5 mm diameter). The ^1^H NMR spectra were recorded using a 600 MHz NMR spectrometer (Bruker, Rheinstetten, Germany).

For NMR data processing, the phase and baseline of all the ^1^HNMR spectra were adjusted by the MATLAB software, and the TSP peak was set at 0 ppm. The water peak (4.5–5.0 ppm) and TSP (−1–0.551 ppm) regions were removed. Then, a multivariate statistical analysis was used to identify the differences among samples.

Metabolic Pathway was analyzed by MetPA (http://www.metaboanalyst.ca accessed on 12 March 2021) to identify the significant pathway as well as to generate the graph. In this study, MetPA on the KEGG database (http://www.genome.jp.kegg/ accessed on 12 March 2021 to identify the significant pathway as well as to generate the graph. In this study, MetP and the Human Metabolomic Database (http://www.hmdb.ca accessed on 12 March 2021) were used.

### 4.12. In Vitro Assay for Inhibition of Aβ Aggregation by Thioflavin-T Assay

The ThT assay for evaluation of the Aβ1-42 self-aggregation process was performed according to the previously described procedure [40]. Briefly, Aβ1-42 was solubilized with 50 mM phosphate buffer (pH 7.4) to give a 250 µM stock solution. The test compounds and curcumin (**4**) were firstly solubilized in DMSO at a concentration of 10 mM. 2 µL of each compound was added to black, opaque 96-well plates in various concentrations (100 µM). After that, Aβ1-42 solution was added to each well giving a final concentration of 10 µM, then gently mixed together. Plates were covered and kept in the dark for 24 h at 37 °C with no agitation. After incubation, 190 µL of 5 µM of ThT in 50 mM glycine/NaOH buffer (pH 8.0) was added to each well and the fluorescence intensities were measured at excitation and emission wavelengths of 446 nm and 490 nm, respectively. The percent inhibition of the test compounds was calculated. All experiments were done in triplicate.

### 4.13. Statistical Analysis

The data obtained from in vitro studies were presented as the mean ± SD. Behavioral and neurochemical data were expressed as the mean ± SEM for each group and analyzed by one-way ANOVA, followed by the post hoc Tukey test. Differences of *p* values < 0.05 were considered to be statistically significant. Data were analyzed using SigmaStat^®^ ver. 3.5 (SYSTAT Software Inc., San Jose, CA, USA). For metabolomic analysis, PCA and OPLS-DA analyses were conducted using SIMCA-P+ version 15 (Umetrics Inc., Umeå, Sweden). The data are mean-centered and Pareto scaled (Par). Moreover, the Statistical Total Correlation Spectroscopy (STOCSY) in the MATLAB software was used to confirm correlations between factors among the chemical shift values. The *p*-value of all models were derived from permutation test (n = 100).

## 5. Conclusions

The current study reveals that *Alternanthera philoxeroides* (Mart.) Griseb is able to cause neurodegenerative changes in NORT and the Y-maze task, respectively. Daily administration with the crude ethanol extract of *A. philoxeroides* was found to improve cognitive deficits-like behavior of the estrogen-deprived mice. Furthermore, *A. philoxeroides* extract also exhibited a capacity to reduce oxidative stress via inhibition of lipid peroxidation in the whole brain and down-regulates neuroinflammatory cytokines (IL-1β, IL-6 and TNF-α) whereas estrogen receptor-mediated facilitation genes (PI3K and AKT) in both frontal cortex and hippocampus were up-regulated similar to that of 17β-estradiol. Moreover, the serum metabolomic analysis was used as a tool to prove that the crude ethanol extract of *A. philoxeroides,* at 500 mg/kg/day, significantly exerts its effect on the mice brain and the metabolites propylene glycol, α-D-glucose, glycerol and betaine were found to be involved in metabolic pathways. The effect of this plant could be due to its flavonoid constituents. Additionally, the flavone constituents showed potent inhibition against Aβ aggregation, a hallmark of Alzheimer’s disease which is one of the Global Burden Diseases mainly affecting elderly people. All these experimental data suggest that *A. philoxeroides* extract could be beneficial for menopausal and ovariectomized women in preventing senile dementia. Moreover, the results also suggest that its bioactive flavone constituents have potential as lead compounds for further development of anti-dementia agents.

## Figures and Tables

**Figure 1 molecules-26-02789-f001:**
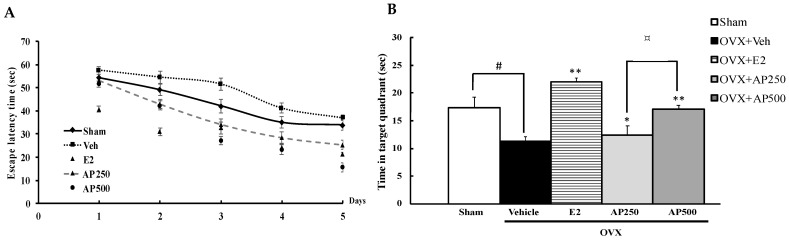
Latency time of training the mice to find the target platform (n = 8–10). Values given were the mean ± SEM of latency, 4 trials/day for 5 days (**A**). The effect of the AP extract on OVX-induced cognitive impairment in the MWMT (**B**). Time in the quadrant where the platform was located was observed on the sixth day. Each column represents the mean ± SEM (n = 8–10). ^#^
*p* < 0.05 vs. the sham-operated group, * *p* < 0.05, ** *p* < 0.001 vs. the OVX group and **^¤^**
*p* < 0.001 vs. the AP extract (post hoc Tukey test).

**Figure 2 molecules-26-02789-f002:**
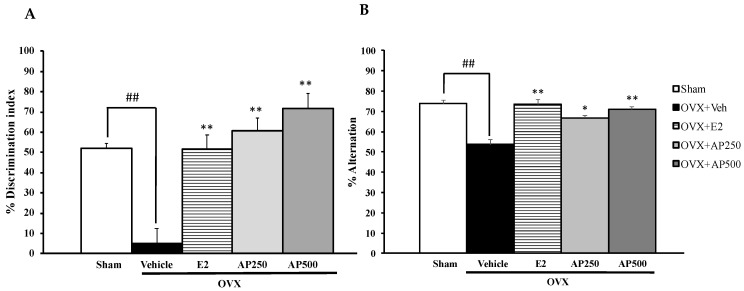
Effect of the AP extract on learning and memory in the sham-operated and OVX mice using the NORT (**A**) and the Y-maze test (**B**). Each column represents the mean ± SEM (n = 8–10). ^##^
*p* < 0.001 vs. the sham-operated group and * *p* < 0.05, ** *p* < 0.001 vs. the OVX group (post hoc Tukey test).

**Figure 3 molecules-26-02789-f003:**
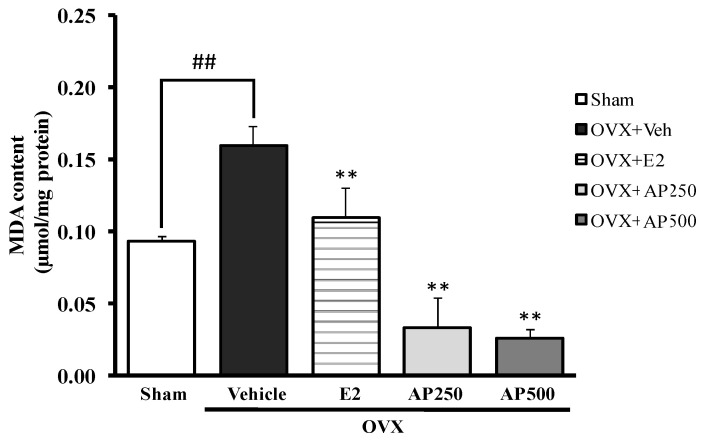
Effect of the AP extract on the OVX-induced oxidative damage in mice whole brain. Values given were the mean ± SEM (n = 6). Significant ANOVA effects were represented by ^##^
*p* < 0.001 vs. the sham-operated group and ** *p* < 0.001 vs. the OVX group (post hoc Tukey test).

**Figure 4 molecules-26-02789-f004:**
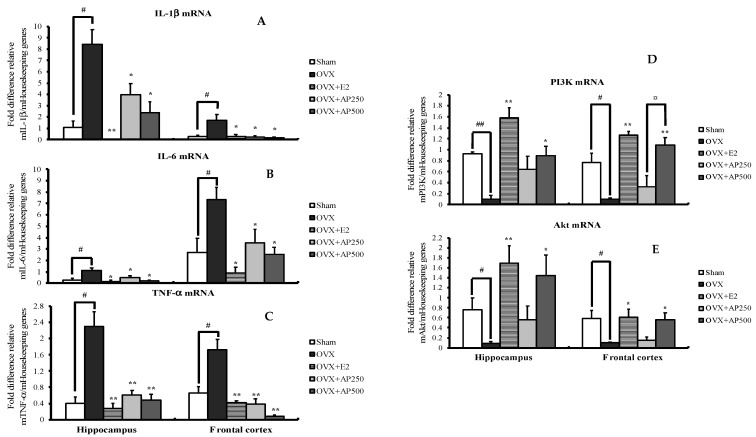
Effect of the AP extract on pro-inflammatory cytokines-related genes (**A**–**C**) and PI3K/AKT pathway-related genes (**D**–**E**) in mRNA expressions of mice hippocampus and frontal cortex. Each column represents the mean ± SEM (n = 5–8). ^##^
*p* < 0.001 vs. the sham-operated group, ^#^
*p* < 0.001 vs. the sham-operated group and * *p* < 0.05, ** *p* < 0.001 vs. the OVX group and **^¤^**
*p* < 0.001 vs. the AP extract (post hoc Tukey test).

**Figure 5 molecules-26-02789-f005:**
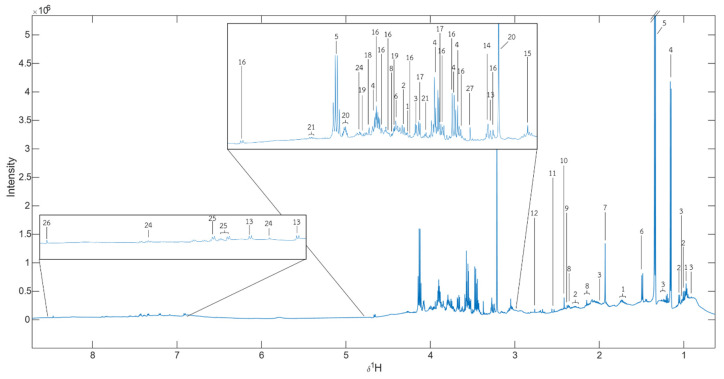
δ_H_ median NMR spectra of all samples. 1 = Leu, 2 = Val, 3= Ile, 4 = Propylene glycol, 5 = Lactate, 6 = Ala, 7 = Unknown 1, 8 = Met, 9 = Pyruvate, 10 = Succinate, 11 = Unknown 2, 12 = Butanedinitrile, 13 = Tyramine, 14 = Betaine, 15 = Creatinine, 16 = α-D-glucose, 17 = Glycerol, 18 = Glycolate, 19 = Ser, 20 = Choline, 21 = Thr, 22 = Fumarate, 23 = Tyr, 24 = His, 25 = Protocatechuic acid, 26 = Formate, 27 = Unknown 3.

**Figure 6 molecules-26-02789-f006:**
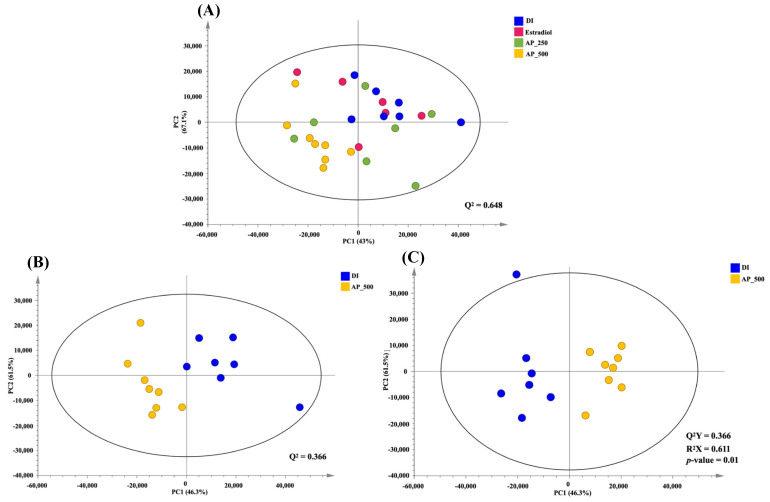
PCA score plot of all interventions; blue = DI water, pink = E2, green = AP extract 250 mg/kg/day, yellow = AP extract 500 mg/kg/day (**A**). PCA score plot between control (DI water) and AP 500 group (**B**). OPLS-DA score plot between control (DI water) and AP 500 (**C**).

**Figure 7 molecules-26-02789-f007:**
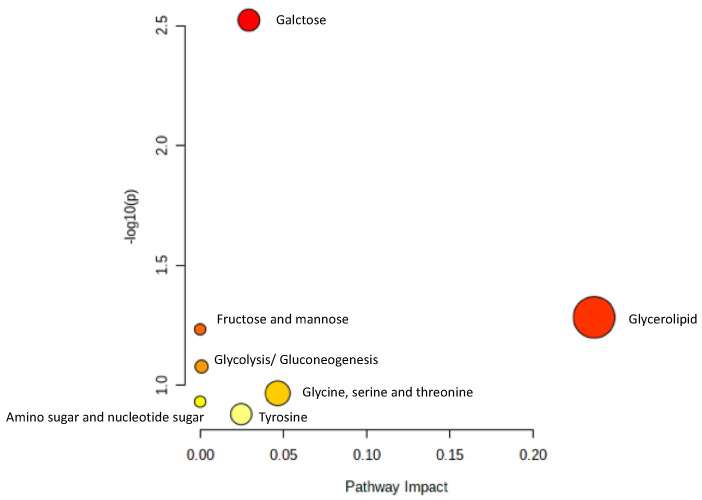
Summary of pathway analysis.

**Figure 8 molecules-26-02789-f008:**
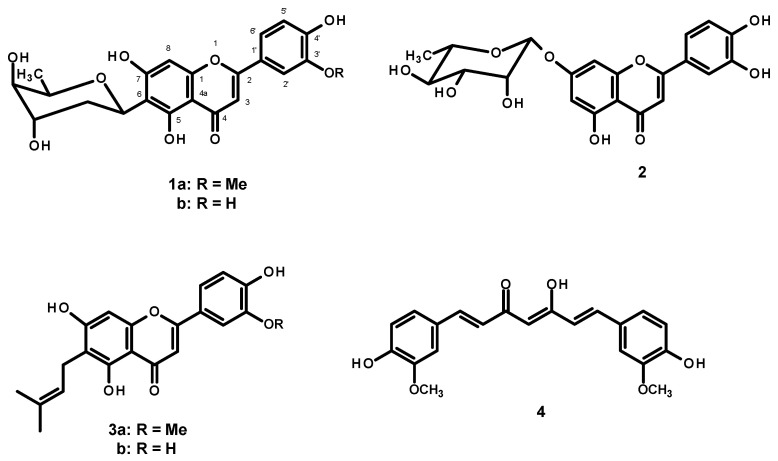
Structures of flavone constituents (**1a**, **1b**, **2**, **3a** and **3b**) of *A. philoxeroides* and curcumin (**4**).

**Figure 9 molecules-26-02789-f009:**
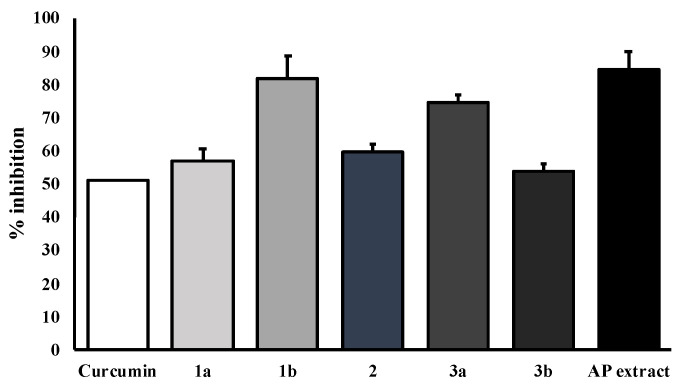
Effects of the AP extract and its flavone constituents on Aβ aggregation at equivalent amounts by ThT assay. The values are presented as mean ± SD (n = 3).

**Figure 10 molecules-26-02789-f010:**
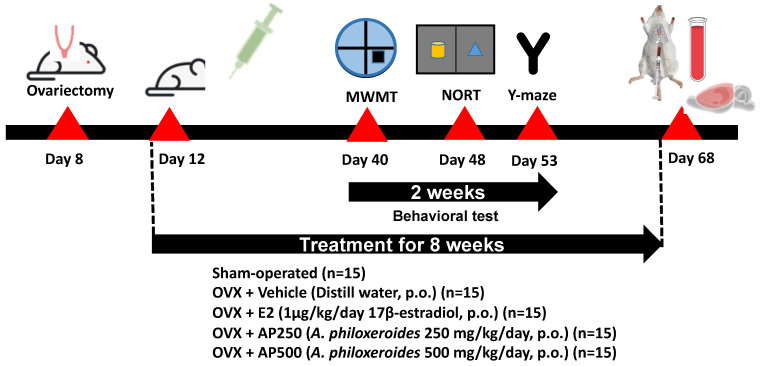
A schematic presentation of animal experimental design.

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
