# Peer review of "Antidementia Effects of Alternanthera philoxeroides in Ovariectomized Mice Supported by NMR-Based Metabolomic Analysis"

_molecules, 2021, doi:10.3390/molecules26092789_

Round 1
Reviewer 1 Report
The manuscript entitled “Antidementia effects of Althernanthera philoxeroides in ovariectomized mice supported by NMR-based metabolomic analysis” (molecules-1199224) by Khamphukdee et al. is a very well written paper presenting the effects of the ethanol crude extract of A. philoxeriodes on dementia-induced by ovariectomy in mice. In my opinion present manuscript will be of interest to many in the field of AD and dementia. However, there are some points that could be improved:Major comments:
- How did you choose the dose of extract given to animals in the study? Did authors perform any preliminary studies about the dose of extract? Are there any dose dependent changes in analyzed parameters?
- How the oral administration (p.o.) of the extract (once daily 0.2 mL per mouse) was performed, with the diet or intragastrically? Intragastrically administration seems to be most efficient and accurate, ensuring the same dose of active compounds to each animal of the experimental groups.
- Distilled water was used as vehicle for ethanol extract. What was the final concentration of ethanol in the AP extract given to the animals? Does ethanol affect oxidative stress parameters (TBARS level), due to its metabolism to 1-hydroxyethyl radical. This should be describe in details in the MM section
- While the study provides interesting and promising results from analysis of many parameters, there is no confirmation if bioactive compounds (flavone constituents) present in the extract are responsible for obtained results. Is this effect of one compound or synergistic effect of a group of constituents? Did the authors determine the levels of flavones in animals treated with AP extract? Authors could make more efforts in the discussion to indicate or explain which compound or a group of compounds could be responsible for observed effects.
- Five metabolites were selected as candidate biomarkers associated with the OVX mice treated with high dose of the AP extract. What was the basis of this selection? Did you try ROC curve approach for evaluating the diagnostic power of a biomarker?
Minor points:
- In the summary, authors could make more efforts to underline effects which are most promising in future perspective of AD management.
- The full explanation for abbreviation of “p.o.” should be given in methodological section.
Author Response
Reviewer #1
We wish to thank reviewer #1 for his/her constructive comments and suggestions which will certainly will help improve a readability of this manuscript.
Reviewer #1: How did you choose the dose of extract given to animals in the study? Did authors perform any preliminary studies about the dose of extract? Are there any dose dependent changes in analyzed parameters?
Reply: The dose of the extract given to animals in the study was based on our previous study which can be found our published paper “Chemical Constituents and Antidepressant-like Effects in Ovariectomized Mice of the Ethanol Extract of Alternanthera philoxeroides” (https://doi.org/10.3390/molecules23092202).The results have shown that the A. philoxeroides extract (250 and 500 mg/kg/day) ameliorated depression-like behaviors in OVX mice via stimulation of CREB- and BDNF-mediated neuroplasticity and neurogenesis in the hippocampus and frontal cortex. So, we did the preliminary studies about the dose of extract.
As to the question if there is any dose-dependent changes in analyzed parameter, our results in this study indicated that there are dose-dependent changes in both of behavioral studies, i. e. the Morris water maze test (as shown in Figure 1A) and PI3K mRNA expression gene in the frontal cortex (as show in Figure 4D). In addition, the Y-maze test (Figure 2B) and Akt mRNA expression results (Figure 4E) also exhibited a different p value between the doses of 250 and 500 mg/kg/day.
Reviewer #1: How the oral administration (p.o.) of the extract (once daily 0.2 mL per mouse) was performed, with the diet or intragastrically? Intragastrically administration seems to be most efficient and accurate, ensuring the same dose of active compounds to each animal of the experimental groups.
Reply: The AP extracts were intragastrically administered to OVX mice using gavage tube. Gavage makes it possible to administer substances directly into the stomach with accurate dosages and reliable timing. We have added the method of administration in the subsection 4.5. Drugs and Drug Administration of the experimental section.
Reviewer #1: Distilled water was used as vehicle for ethanol extract. What was the final concentration of ethanol in the AP extract given to the animals? Does ethanol affect oxidative
stress parameters (TBARS level), due to its metabolism to 1-hydroxyethyl radical. This should be described in details in the MM section.
Reply: Powdered whole plants of A. philoxeroides was Soxhlet-extracted with EtOH (3 x 10 L) at 50ºC for 1 hr. The ethanol solution was evaporated under reduced pressure at 40-45 ºC and then the moisture was removed by Freeze-drying. This process of extraction gives ethanol-free crude extract. Therefore, the AP extract used in this study will not affect any oxidative stress parameters (TBARS level) in mice brain. We have added a sentence that “the crude ethanol extract was freeze-dried before giving to the animals” in Sub-section 4.1. Plant Material, Preparation of Plant Extract and Isolation of Flavone Constituents
Reviewer #1: While the study provides interesting and promising results from analysis of many parameters, there is no confirmation if bioactive compounds (flavone constituents) present in the extract are responsible for obtained results. Is this effect of one compound or synergistic effect of a group of constituents? Did the authors determine the levels of flavones in animals treated with AP extract? Authors could make more efforts in the discussion to indicate or explain which compound or a group of compounds could be responsible for observed effects.
Reply: In order to ensure that flavonoids could be responsible for the antidementia effect, we designed to investigate the b-amyloid aggregation inhibitory effect of the crude ethanol extract of A. philoxeroides and its flavone constituents, alternanthin (1a), alternanthin B (1b), chrysoeriol 7-rhamnoside (2), torosaflavone E (3a), and demethyltorosaflavone D (3b) in the present study. The results showed that the AP extract and these five flavone derivatives exhibited a pontent inhibitory effect on b-amyloid aggregation. Aβ aggregation into plaques leads to neurotoxicity and dementia through common cytopathic effects that contribute to the pathogenesis of AD. In addition, we have also previously determined the effect of these flavone derivatives on the function of monoamine oxidase (MAO) enzymes ((https://doi.org/10.3390/molecules23092202). The results demonstrated that these five flavones exerted the MAO-A and MAO-B inhibitory effects. Alternanthin (1a) showed more potent inhibitory effect on MAO-A when compared with clorgyline (selective MAO-A inhibitor) and alternanthin B (1b) exhibited more potent inhibitory effect on MAO-B when compared with deprenyl (selective MAO-B inhibitor). It is well established that MAO plays an important role in several key pathophysiological mechanisms in AD and neurodegenerative diseases. MAO-B has been proposed as a biomarker, whereas activated MAO-B leads to cognitive dysfunction, destroys cholinergic neurons, causes disorder of the cholinergic system and contributes to the formation of amyloid plaques. MAO inhibitors improve cognitive deficits and reverse Aβ pathology by modulating proteolytic cleavage of amyloid precursor protein and decreasing Aβ protein fragments. Taken together, these findings can confirm the pharmacological activity of a group of flavone derivatives that responsible for the antidementia effect of AP extract. In order to improve a readability and clarification, we have added this statement in the discussion (pages 434-439)
Reviewer #1: Did the authors determine the levels of flavones in animals treated with AP extract?
Reply: Concerning the question if we have determined the levels of flavones in animals treated with theAP extract, our answer is we did not do it in the present study. The reason is that it was not the objective of this work. In this work we used the metabolomics analysis as a tool to determine the effect of the AP extract. However, we consider the question raised by reviewer #1 is interesting for our further investigation. This also will need an appropriate analytical method as the concentration of the flavones in the crude extract is very low. Therefore, a very sensitive method to detect the levels of these compounds in the serum is required
Reviewer #1: Authors could make more efforts in the discussion to indicate or explain which compound or a group of compounds could be responsible for observed effects.
Reply: We have added the sentences that explain about the pharmacological activity of the group of compounds in the discussion part.
Reviewer #1: Five metabolites were selected as candidate biomarkers associated with the
OVX mice treated with high dose of the AP extract. What was the basis of this selection?
Reply: We wish to thank reviewer #1 for raising this question. The metabolomics experiment performed in the current study follows the typical pipeline that was previously published (https://doi.org/10.1038/nprot.2007.376). The multivariate statistical analysis was used to investigate the differences in all intervention group. A principal component analysis (PCA) was first applied to the data to identify outliers and visualize the underlying trends as well as showing the variation in the matrix data (Figure 6A-6B). As can be seen from the PCA plot, there were noticeable overlaps between the two classes of samples, indicating that the samples could not be separated well. As a result, a clear deviation was observed between the control group (DI water) and OVX mice treated with high dose of the AP extract (500 mg/kg/day), indicating that OVX mice treated with high dose of the AP extract (500 mg/kg/day) presented the best performance against the deviations induced by OVX. Next, orthogonal projections on latent structure-discriminant analysis (OPLS-DA) was introduced for discrimination and potential markers identification. Our results shown a clear separation between the two groups in Figure 6C. 1HNMR spectra of differential metabolites between these groups exhibited a significant difference (p<0.05) only for tyramine, betaine, propylene glycol, α-D-glucose and glycerol, indicating a higher correlation in high dose of the AP extract (Table S16). Therefore, five differential metabolites were elected as candidate biomarkers associated with the OVX mice treated with high dose of the AP extract.
https://doi.org/10.1038/nprot.2007.376
Reviewer #1: Did you try ROC curve approach for evaluating the diagnostic power of a biomarker?
Reply: We did not try the ROC curve to determine the diagnostic power of a biomarker for two reasons. First, although in Receiver-Operating Characteristic (ROC) curve analysis, the area under the curve is a useful tool for evaluating the performance of diagnostic power of a biomarker or for comparison of competing biomarker (mostly in clinical study) by generally the accuracy of a statistical model (e.g., logistic regression, linear discriminant analysis) (https://doi.org/10.1161/CIRCULATIONAHA.105.594929), the AUC and the maximum of the Youden index, J are used both for the predictive of an optimal cutoff point that can be used to assess a single biomarker and for comparison of two biomarkers. That is why it is not appropriate for our study because a statistical model of ROC can classify subjects into 1 of 2 categories, for example, diseased or non-diseased (https://doi.org/10.1007/s10182-020-00371-8), whereas our samples come from four interventions that need to determine which one is the most effective for OVX condition. Secondly, in this study we focus on serum metabolites analysis, up-to-date technique to prove that this effect is due to the AP extract or not? For the basic of metabolites selection as a biomarker by 1H-NMR-Metabolomic analysis can be described as firstly, untargeted metabolomics was determined simultaneously to detect as many metabolites as possible in the mice serum in a given sample such as deionized water (DI), 17β-estradiol (E2) and the AP extract (250 and 500 mg/kg/day) by using 1H NMR to identify 27 metabolites as a metabolic profiling of serum sample (Figure 5). Lastly, Metabolic Pathway Analysis (MetPA) was used to perform the most relevant metabolites (tyramine, betaine, propylene glycol, α-D-glucose and glycerol) and metabolic pathway by MetPA (http://www.metaboanalyst.ca) in conjunction with Human Metabolome Database. The AP extract was found to interfere with the galactose metabolism pathway. It is important to point out that decreased glucose supply for brain cells is related to the onset of AD (we have already expressed in the manuscript). From all of this explanation can be summarized that 1HNMR metabolomic analysis is an up-to-date technique to determine a metabolite from biological sample of animal treated with medicinal plant in the in vivo study. Moreover, this technique also reveals the pathway based on metabolome database which is related to disease condition and the treatment.
Reviewer #1: In the summary, authors could make more efforts to underline effects which
are most promising in future perspective of AD management.
Reply: We have added a conclusive statement in the conclusion.
Reviewer #1: The full explanation for abbreviation of “p.o.” should be given in
methodological section.
Reply: We have given the full explanation for abbreviation of “p.o.” in 4.5. Drugs and Drug Administration
Reviewer 2 Report
The paper entitled ‘Antidementia effects of althernanthera philoxeroides in ovariectomized mice supported by NMR-based metabolomic analysis’ prepared by Khamphukdee et al. presents interesting and valuable studies results based on positive effect of crude extract from selected plant on dementia, especially women. The aspect of estrogen in AD development is often marginalize nevertheless it is known that the hormone can significantly influence on neurodegeneration. Authors decided to use various methods (in vitro, in vivo) which allowed to determine antioxidant activity and behavioral impairment counteraction in mice model. The obtained results are presented in form of tables and figures. Authors discussed the results in detail using up-to-date references. In my opinion the article is interesting, methods are well-addapted, results and discussion are well-presented and readable. The following points should be corrected/improvement:
- Methods: DPPH assay the absorbance was measured after 30min from reaction initiation. In my opinion, that is not exactly good solution, because change of absorbance value is very important for antioxidant evaluation – even weak free radical scavengers can scavenge free radicals after 30 min what can indicate on their activity comparable with compounds which are able to scavenge free radicals after few minutes after reaction initiation.
- Please correct stylistic and interpunction errors in the manuscript
Author Response
Reviewer #2
We wish to thank reviewer #2 for his/her appreciation of the manuscript as well as his/her comments and suggestions.
Reviewer #2: Methods: DPPH assay the absorbance was measured after 30 min from reaction initiation. In my opinion, that is not exactly good solution, because change of absorbance value is very important for antioxidant evaluation – even weak free radical scavengers can scavenge free radicals after 30 min what can indicate on their activity comparable with compounds which are able to scavenge free radicals after few minutes after reaction initiation.
Reply: The DPPH radical scavenging effect was evaluated according to the methods of Othman and co-worker who determined antioxidant activity of Alternanthera sessilis. The important point that reviewer #2 has raised is the DPPH· can be reacted with even weak antioxidants and also strong antioxidants so the rate of reaction between DPPH· and substrates varies with time. A reaction time as short as 5 to 10 min has been reported but in most analyses, 30 min is an appropriate period to reach a steady state (https://doi.org/10.1016/j.jff.2015.12.014). This condition is published by several research groups. However, as no single method can give an accurate results and that’s why we performed the ABTS assay to evaluate antioxidant activity of our plant extract based on its correlated mechanism via electron transfer (ET). The difference between the two methods is that the ABTS assay is applicable to both hydrophilic and lipophilic antioxidant systems, whereas the DPPH assay is more suitable for the hydrophobic system (DOI. 10.2174/1573411013666170118111516).
Reviewer #2. Please correct stylistic and interpunction errors in the manuscript
Reply: We have checked the stylistic and interpunction errors in the manuscript as requested.
Round 2
Reviewer 1 Report
The authors did a great job in revising the manuscript and provided sufficient explanation to all comments. In my opinion, the manuscript in present form can be accepted for publication in Molecules.